# Recent Progress on Circular RNAs in the Development of Skeletal Muscle and Adipose Tissues of Farm Animals

**DOI:** 10.3390/biom13020314

**Published:** 2023-02-07

**Authors:** Shanying Yan, Yangli Pei, Jiju Li, Zhonglin Tang, Yalan Yang

**Affiliations:** 1Guangdong Provincial Key Laboratory of Animal Molecular Design and Precise Breeding, Key Laboratory of Animal Molecular Design and Precise Breeding of Guangdong Higher Education Institutes, School of Life Science and Engineering, Foshan University, Foshan 528231, China; 2Shenzhen Branch, Guangdong Laboratory for Lingnan Modern Agriculture, Agricultural Genomics Institute at Shenzhen, Chinese Academy of Agricultural Sciences, Shenzhen 518124, China; 3Kunpeng Institute of Modern Agriculture at Foshan, Foshan 528226, China

**Keywords:** circRNAs, animal, skeletal muscle, adipose, development and growth

## Abstract

Circular RNAs (circRNAs) are a highly conserved and specifically expressed novel class of covalently closed non-coding RNAs. CircRNAs can function as miRNA sponges, protein scaffolds, and regulatory factors, and play various roles in development and other biological processes in mammals. With the rapid development of high-throughput sequencing technology, thousands of circRNAs have been discovered in farm animals; some reportedly play vital roles in skeletal muscle and adipose development. These are critical factors affecting meat yield and quality. In this review, we have highlighted the recent advances in circRNA-related studies of skeletal muscle and adipose in farm animals. We have also described the biogenesis, properties, and biological functions of circRNAs. Furthermore, we have comprehensively summarized the functions and regulatory mechanisms of circRNAs in skeletal muscle and adipose development in farm animals and their effects on economic traits such as meat yield and quality. Finally, we propose that circRNAs are putative novel targets to improve meat yield and quality traits during animal breeding.

## 1. Introduction

Skeletal muscle accounts for the largest proportion (approximately 30–50%) of mammalian body mass [1,2,3]. Adipose tissue is the largest energy storage and secretory organ required to maintain physiological functions and metabolic balance [4,5]. Therefore, the development of skeletal muscle and adipose is complex and regulated by multiple physiological factors and biological pathways. In farm animals, skeletal muscle and adipose development are directly associated with several vital economic traits, such as meat yield and quality [6,7,8]. Intramuscular fat content is the predominant factor determining the meat quality of farm animals. With the rapid development of molecular breeding and functional genomics in farm animals, many candidate and marker genes associated with pork quality have been identified and used as essential selection criteria for breeding programs [9,10]. Examples include insulin-like growth factor 1 receptor (IGF1R) [11], and protein kinase AMP-activated non-catalytic subunit gamma 3 (PRKAG3) [12]. Beyond protein-coding genes, non-coding RNAs (ncRNAs) also play crucial roles in the development of skeletal muscle and adipose through various regulatory mechanisms; therefore, they are also closely associated with meat yield and quality traits [13,14,15,16]. 

Circular RNAs (circRNAs) are a class of covalently closed ncRNAs with no 5′–3′ polarity and polyadenylation tail (polyA) [17]. In 1976, the first circRNA was discovered in viroid RNA viruses and confirmed by nucleotide sequencing [18,19]. In 1979, circRNAs were identified in eukaryotes [20]. In 1993, researchers further understood the biosynthesis of circRNAs, which was considered an erroneously spliced mRNA product. They demonstrated that the human ets-1 transcript was a circular RNA molecule containing only exons whose the normal order of exons was inverted. This is the first case of pre-mRNA processing into a circular transcript [21]. CircRNAs are ubiquitous in vertebrates and exhibit spatial-temporal-specific expression patterns [22,23,24,25,26,27,28]. In addition, circRNAs are highly conserved, stable, and resistant to exonucleases. In 2013, the first functional study of a naturally expressed circRNA, CDR1as (also known as ciRS-7), which could act as a miR-7 sponge, was reported [29]. Since then, the identification, function, and mechanistic analysis of circRNAs has become a research hotspot in the RNA community. The functions of circRNAs in regulating gene expression [30,31], protein translation [32], regulation of mRNA stability [33], and biomarker activity [34] have been widely reported.

Continued innovation in high-throughput sequencing technology and bioinformatics tools have facilitated the identification and functional characterization of circRNAs in humans, animals, plants, and fungi. Increasing evidence suggests that circRNAs play critical roles in the development and growth of skeletal muscle and adipose tissues [35,36]. For example, we recently reported that circFgfr2, a conserved circRNA among species produced from the fibroblast growth factor receptor 2 (*Fgfr2*) gene, regulates myogenesis and muscle regeneration via a miR-133/Map3k20/JNK/Klf4 auto-regulatory feedback loop [37]. Additionally, circTshz2-1 and circArhgap5-2 are essential for adipogenesis [38]. 

Thousands of circRNAs have been identified in the transcriptome of farm animals. Functional studies have suggested that circRNAs are essential for skeletal muscle development and adipose deposition and are associated with meat yield and quality. In this review, we have summarized the biogenesis, properties, and biological functions of circRNAs, and highlighted recent advances in the functions and regulatory mechanisms of circRNAs during the development of skeletal muscle and adipose tissues in farm animals. We have also explored their associations with traits of meat yield and quality. CircRNAs could serve as novel targets to improve meat yield and quality in animal breeding. 

## 2. Biogenesis and Properties of CircRNAs

CircRNAs are generated by a “back-splicing” process. The biogenesis of circRNAs reportedly involves several mechanisms, such as reverse splicing, *cis*-regulatory elements, and RNA-binding protein (RBP)-driven circularization [39,40]. Most circRNAs in eukaryotes are transcribed from pre-mRNAs [41]. They can arise from different genomic regions, such as exons (exonic circRNAs), introns (intronic circRNAs, ciRNA), and exon–intron (exon–intron circRNAs, EIciRNA) [42,43]. The spliceosomes join the downstream 5′ and upstream 3′ splicing sites through back-splicing (Figure 1a). A single gene can produce multiple circRNAs with different lengths and sequences by alternative back-splicing, but these circRNAs share the same back-splicing site [44]. In addition, multiple RNA pairs in introns can compete with each other for alternate cyclization to form different circRNAs. For example, DNMT3B and XPO1 can generate multiple circular exons by alternate circularization [45]. Back-splicing combined with transcription could increase the biogenesis of circRNAs by inhibiting the co-transcription 3′ terminal processing [39,46,47]. Back-splicing can compete with linear RNA splicing by direct reverse splicing and nesting intermediates to produce a mature circRNA [48,49,50,51]. 

In the *cis*-regulatory model, *cis*-elements can reduce the spatial steric hindrance of back-splicing owing to the proximity of the splicing sites brought about by RNA pairing between introns on both sides of the exon (Figure 1b) [45,46]. Moreover, the widely distributed Alu elements in long introns may allow multiple RNA pairings and enhance circRNA production. For example, circHIPK3 was derived from exon2 of the homeodomain-interacting protein kinase 3 (*HIPK3*) gene. The flanking introns of *HIPK3* exon2 exhibited highly complementary Alu repeats, with 28 short interspersed nuclear elements (SINEs) in the intron upstream of *HIPK3* exon2 and 51 SINEs downstream of exon3, which are required for the circularization of circHIPK3 [52].

In addition, two RNA-binding protein modes regulate the back-splicing sites (Figure 1c): It stabilizes the transiently formed intronic RNA pairs flanking circRNA-forming exons to facilitate back-splicing [51]. Several other factors, such as NF90/NF110 [53] and N6-methylation of adenosine [54], also reportedly coordinate the biogenesis of circRNAs.

## 3. Regulatory Mechanisms of CircRNAs

CircRNAs are ubiquitously expressed in various tissues and highly enriched in the brain and reproductive organs. CircRNAs reportedly play crucial roles in regulating development, aging, and disease in humans and other mammals through various regulatory mechanisms, such as acting as miRNA/protein sponges and transcriptional regulators, interacting with proteins, and translating into proteins. In addition, circRNAs can directly regulate gene expression or interact with other molecules to function across myriad biological contexts.

### 3.1. miRNA Sponges

CircRNAs most commonly exert regulatory effects by acting as miRNA sponges. CircRNAs are predominantly enriched in the cytoplasm, although they are generated in the nucleus. Recent studies have also suggested that some circRNAs are mainly enriched in the nucleus or simultaneously expressed in the cytoplasm and nucleus. Most cytoplasmic circRNAs are intron-free [51,55] and act as competitive endogenous RNA (ceRNAs) that function as miRNA sponges [51]. Notably, when circRNAs function as miRNA sponges, the stoichiometric relationship between circRNA-binding sites and miRNA targets is particularly vital because a circRNA may be targeted by one miRNA at multiple loci or by various miRNAs. For example, circRNA CDR1as has approximately 70 binding sites for miR-7 [13], which can upregulate the expression of insulin-like growth factor 1 receptor (IGF1R) by acting as a miR-7 sponge (Figure 2a), thereby activating muscle differentiation [56]. CircHIPK3 has 18 binding sites with nine miRNAs [52], which directly bind miR-124 to regulate the growth of human cells and absorb miR-326 in the cytoplasm to regulate the proliferation, migration, and apoptosis of smooth muscle cells [57]. circHIPK3 can promote myoblast proliferation and differentiation by sponging miR-7 and increasing TCF12 expression [35]. 

### 3.2. Interactions with Proteins

CircRNAs also function as protein sponges and scaffolds and have important effects on multiple physiological pathways [58]. Some circRNAs have specific protein-binding sites [43], which can regulate the subcellular localization of specific proteins [59], recruit proteins to specific locations [60], facilitate protein–protein interactions, and assist the assembly of protein complexes [61]. One circRNA can bind to one protein or interact with multiple proteins [62]. For example, circFOXK2 can directly interact with 94 proteins, such as ANK1, GDNF, and PAX6 [63]. Circ-Foxo3 forms ternary complexes with cyclin-dependent kinase inhibitor 1 (p21) and cyclin-dependent kinase 2 (CDK2) (Figure 2b) [61]. Furthermore, circ-Foxo3 can act by binding to proteins in related signaling pathways. circ-Foxo3 increases the expression of Foxo3 at the protein level but represses the protein level of tumor protein 53 (p53). By binding to both proteins, circ-Foxo3 can promote mouse double minute 2 (MDM2)-induced ubiquitination and degradation of p53, leading to an overall decrease in p53 [64]. Meanwhile, circFoxo3, located in the cytoplasm, is associated with mouse fibroblast senescence by interacting with anti-aging protein inhibitors of DNA binding-1 (ID-1) and E2F transcription factor 1 (E2F1) as well as anti-stress proteins FAK (Recombinant focal adhesion kinase) and HIF1α (Recombinant hypoxia inducible factor 1 alpha). This leads to accelerated cellular aging [65]. 

Meanwhile, circRNAs affect the subcellular localization of proteins. CNEACR is a cardiac necrosis-associated circRNA which binds directly to histone deacetylase 7 (HDAC7) in the cytoplasm and interferes with its nuclear entry. This leads to reduced transcriptional inhibition of forkhead box A2 (Foxa2), thereby upregulating receptor-interacting protein kinase 3 (Ripk3) gene to affect myocardial necrosis [66]. In addition, circMb1 can act as a protein sponge. CircMb1 has a strong interaction with MBL protein. When MBL protein is excessive, it can promote the production of circMb1 to reduce its mRNA production, and circMb1 can absorb excess MBL protein by binding to it [31]. CircNDUFB2 acts as a scaffold to enhance the interaction between three motif protein 25 (TRIM25) and insulin-like growth factor 2 mRNA binding proteins (IGF2BPs), forming a TRIM25/CircNDUFB2/IGF2BPs ternary complex. This complex promotes ubiquitination and degradation of IGF2BPs, thus affecting the progression of non-small cell lung cancer [67]. Many other circRNAs, such as CircRNA-CREIT [68], CircEIF3H [69], and CDOPEY2 [70], have also been reported to function as protein scaffolds.

### 3.3. Translation Potential of CircRNAs

As circRNAs lack 5′ capped and 3′ polyadenylated ends, it was thought that they could not be translated. However, recent studies have suggested that a subset of circRNAs can be translated into proteins/peptides via two main cap-independent mechanisms (Figure 2c): internal ribosome entry site (IRES)-dependent [71] and m6A-mediated translation [72]. For example, circ-ZNF609 could be translated into a protein in a splicing-dependent and cap-independent manner to regulate myogenesis [73]. IRESs are present in viral or cellular genes, which promote ribosome assembly and initiate translation by recruiting different trans-acting factors. Most IRESs are regulated by IRES *trans*-acting factors (ITAFs). Almost all ITAFs are RNA binding proteins, such as PABPC1, hnRNP U, ELAVL1 (HuR), and hnRNP A1, recruited by IRES elements to facilitate the ribosome assembly onto pre-mRNA, and thus promote the translation of circRNAs [71,74,75]. Meanwhile, the complementarity of 18s RNA and structural RNA elements located on IRES are particularly important for the translation of circRNAs [76]. In comparison, 18s RNA was shown to match the short *cis*-element in IRES to recruit ribosomes to initiate translation [71].

Several factors, including RNA N6-methyladenine (m6A) modification, heat shock, polyadenylate sequence, intron-mediated enhancement (IME), and acute fasting, affect the translation of circRNAs [77]. For example, m6A is produced by the deposition of adenosine methyltransferase complexes (METTL3 and METTL14), possibly promoting the translation of circRNAs from reporter genes and endogenous loci [72]. Furthermore, it is the most abundant internal modifications in coding and non-coding RNA polymerase II transcripts, mainly responsible for the translation of hybrid circRNAs, which contain a large number of m6A modifications sufficient to drive protein translation in a cap-independent manner, including the m6A reader YTHDF3 and the translation initiation factors eIF4G2 and eIF3A [72,78]. For example, m6A regulates the translation of circ-ZNF609 by recognizing YTHDF3 and eIF4G2 [78]. Additionally, m6A demethylase fat and obesity-related (FTO) protein can reduce the translation efficiency of circRNAs [72,78]. 

### 3.4. Transcriptional Regulators 

CircRNAs can be transcribed through specific RNA–RNA interactions between U-snRNAs, thereby enhancing the delivery of their parent genes in *cis*-elements. EIciRNA is a new class of circRNAs containing introns and exons, such as circEIF3J and circPAIP2. EIciRNA and U1 snRNP (U1A and U1C) occupy approximately 300 bp upstream of the transcription initiation site of the parent gene. EIciRNA, U1 snRNP, and Pol II may interact in the promoter region of the parent gene (Figure 2d). Li et al. reported that EIciRNAs might maintain factors such as U1 snRNP through RNA–RNA interaction between U1 snRNP and EIciRNA. The EIciRNA-U1 snRNP complex further interacts with the Pol II transcription complex at the promoter to enhance parental gene expression [79]. Simultaneously, circRNAs can coordinate with *trans*-acting factors to promote their biogenesis [80]. In addition, circular intron RNAs (ciRNAs) readily form DNA:RNA hybrids (R-loops) owing to their high GC content and can affect transcription through Pol II. Among them, ci-ankrd52 has a strong R-loop formation ability and can be used as a positive regulator of Pol II transcription to affect gene transcription [30,81]. Meanwhile, circRNAs can regulate gene transcription by binding to the gene promoter. For example, circIPO11 can recruit TOP1 to the promoter of GLI1, triggering its transcription and activating Hedgehog signaling [82]. CircRNA-FECR1, composed of FLI1 exon 4-2-3, can bind to the FLI1 promoter to activate gene transcription [60]. CircHipk2 regulates the expression of Hipk2 and circHipk2 by binding to transcription factor Sp1 [83].

## 4. Methodologies for CircRNA Identification

### 4.1. CircRNAs Identification by Bioinformatic Methods

Recent advances in sequencing technologies and bioinformatics tools have greatly facilitated the discovery and analysis of circRNAs. However, the library types of RNA-seq significantly affect the abundance and number of circRNAs [84]. Currently, RNA-seq libraries are typically either poly(A)-selected or depleted of rRNA before library preparation. In theory, poly(A)-selected RNAs should not contain circRNAs. However, a tiny proportion of circRNAs still exist, because selection is not absolutely accurate. CircRNAs are retained in rRNA-depleted libraries and are enriched in libraries treated with RNase R to digest linear RNA. Beyond RNA-seq data, circRNAs could be identified from CLIP-seq, Ribo-seq and miRNA-seq data [85,86,87]. 

Many bioinformatics tools have been developed using different strategies or algorithms. The BSJ read is a typical molecular signature of a true circRNA. Most of the identification tools identify circRNAs by recognizing the BSJ reads from the RNA-seq data, such as Find_circ [88], CIRI [89] and CIRCexplorer [90]. Some tools apply machine learning algorithms to predict circRNAs. They train a classification model by using heterogeneous features of real circRNAs, such as structure and sequence motifs, Alu and tandem repeats, and SNP densities. For example, PredcircRNA used a multiple kernel learning algorithm to distinguish circRNA from other ncRNAs [91]. Meanwhile, some tools integrate different circRNA identification tools to reduce the false-positive rate, such as CirComPara [92], which detect, quantify and annotate circRNAs from RNA-seq data using in parallel four different methods (CIRCexplorer, CIRI, find_circ, and segemehl [93]). 

### 4.2. Identification of CircRNAs by Wet-Lab Experiments

To identify circRNAs and distinguish them from linear RNAs, northern blotting using a probe targeting BSJ is the gold-standard technique for validating circRNAs; however, northern blotting is labor-intensive and time-consuming, requires a large amount of RNA, and often needs to use radioactively labeled probes [94,95,96]. Thus, divergent primers flanking the BSJ sequences are usually designed for RT-PCR to amplify circRNAs. Sanger sequencing was performed to validate the sequence of BSJ [97,98]. Meanwhile, another characteristic of circRNAs is that they are naturally resistant to RNase R. At the same time, most of the linear RNA will be degraded by RNase R. The RNase R digestion experiment can further determine that circRNAs are closed non-coding RNAs. However, it should be noted that circRNA will also be degraded under prolonged RNase R digestion, so the digestion time needs to be controlled [99]. In addition, divergent and convergent primers can be designed to amplify circRNA using genomic DNA (gDNA) and cDNA, respectively. Since only single-stranded cDNA can make divergent primers amplify circRNA, it can be found by gel electrophoresis that circRNA cannot be amplified in gDNA. In contrast, circRNA can be amplified in cDNA [37,100].

## 5. Identification and Functions of CircRNAs in Skeletal Muscle of Farm Animals

### 5.1. Pigs

Pigs are an important protein source for humans and an ideal animal model widely used in human health and disease research. Several research teams, including ours, have studied the molecular regulation of skeletal muscle through processes such as DNA/RNA methylation [101,102], chromatin accessibility [103], and coding and non-coding RNA expression [104], at various developmental stages for several breeds at multi-omics levels. At the circRNA level, we performed circRNA expression profiling in different tissues and constructed the first public circRNA database in pigs. Using Ribo-Zero strand-specific RNA-seq data, we identified 5934 circRNAs in nine organs. After comparing circRNA expression differences among the three postnatal stages (D0, D30, and D240) in the skeletal muscle, we identified 149 circRNAs potentially associated with muscle growth [105]. Recently, our group used strand-specific RNA-seq to profile the circRNAome landscape of skeletal muscles across 27 developmental stages (15 prenatal and 12 postnatal) in Landrace pigs. We also identified 52,918 high-confidence circRNAs in pigs, including 2916 that are conserved across humans, mice, and pigs. Four conserved circRNAs (circFgfr2, circQrich1, circMettl9, and circCamta1) are differentially expressed during pig skeletal muscle development and myoblast differentiation. CircFgfr2 is produced by exons 3–6 of the fibroblast growth factor receptor 2 (*Fgfr2*) gene. Notably, functional and mechanistic analyses suggested that circFgfr2 inhibits myoblast proliferation and promotes differentiation and skeletal muscle regeneration through miR-133/Map3k20/JNK/Klf4 autoregulatory feedback loops (Table 1) [37]. CircRNA analysis of embryonic muscle in Duroc pigs at 33-, 65-, and 90-days post-coitus (dpc) suggested that among the 7968 circRNAs, most were specifically expressed at the early stage. The key myogenesis regulators, PITX2, FGF2, CTNNB1, HOMER1, and HMG20B, were regulated by multiple circRNAs. CircTUT7 acted as a sponge for miR-30a-3p to regulate the expression of HMG20B [106]. Many studies have also compared the differences in the expression levels of circRNAs in the skeletal muscle of different pig breeds [87,107,108]. For example, analysis of differentially expressed circRNAs (DE-circRNAs) in the skeletal muscle of Large White (Western commercial breed) and Mashen (local Chinese breed) pigs suggested that circ_0015885/miR-23b/SESN3 axis might play a vital role in skeletal muscle growth and adipose deposition [107]. 

There are four types of muscle fibers: slow-twitch oxidative, fast-twitch glycolytic, fast-twitch oxidative-glycolytic, and intermediate muscle fibers [109]. The ratio of different muscle fiber types is highly related to meat quality traits in farm animals. Li et al. analyzed the expression of circRNAs in the fast-twitch biceps femoris and slow-twitch soleus muscles of pigs to identify circRNAs that are potentially involved in the transformation of muscle fiber types. They identified 243 DE-circRNAs between these two types of skeletal muscles, which are potential candidates for the regulating skeletal muscle fiber conversion [42]. Cao et al. performed RNA-seq on the *longissimus dorsi* (fast muscle) and soleus (slow muscle). Among the 40,757 circRNAs identified, 181 were significantly differentially expressed. Many DE-circRNAs are involved in metabolism, and AMPK, FOXO, and PI3K-AKT signaling pathways. Notably, injecting circMYLK4-AAV into piglets to overexpress circMYLK4 in muscles could significantly increase the expression of slow muscle marker genes at mRNA and protein levels, suggesting that circMYLK4 is a regulator of skeletal muscle fiber type transitions [110]. These findings strongly suggest that circRNAs play crucial roles in pigs’ skeletal muscle development and fiber-type transition 

### 5.2. Bovines

Cattle are an important source of milk, meat, and draft power. The Hong Chen group examined circRNA expression profiles of bovine skeletal muscle at embryonic (90 days) and adult (24 months) stages, constructed a circRNA-associated ceRNA network for skeletal muscle development, and identified 216 DE-circRNAs between two developmental stages [111,112,113,114]. The functions and regulatory mechanisms of several candidate DE-circRNAs, such as circLMO7, circHUWE1, circSNX29, and circFUT10, were explored (Table 1). They demonstrated that circLMO7 could regulate cell differentiation and survival as sponges for miR-378a-3p in myoblasts [111]. CircHUWE1 promoted myoblast proliferation, reduced apoptosis and differentiation by sponging miR-29b, and eliminated the inhibition of AKT3, thereby indirectly activating the AKT signaling pathway to affect the activity of many vital metabolic targets in myoblasts [112]. CircSNX29 could promote bovine myoblast differentiation by acting as an endogenous miR-744 sponge to activate the Wnt5a/Ca^2+^/CaMKIId pathway [113]. CircFUT10 inhibited myoblast proliferation and induced cell differentiation, with three miR-133a binding sites, and regulated cell differentiation and survival by sponging miR-133a in bovine myoblasts [114]. In addition, circTTN could significantly promote the proliferation and differentiation of bovine primary myoblasts and play its function by competing with miR-432 to activate IGF2/PI3K/AKT signaling pathway [115]. CircRILPL1 acted as a miR-145 sponge to regulate the expression of IGF1R and rescue the inhibitory effect of miR-145 on the PI3K/AKT signaling pathway, thereby promoting myoblast growth [116]. CircFGFR4 could increase myoblast differentiation and decrease apoptosis [117], and circINSR promoted myoblast development [8]. These results highlight the roles of circRNAs in bovine skeletal muscle development and may serve as candidate biomarkers to improve meat production traits in cattle.

In addition, circRNAs play a critical role in muscle regeneration in cattle. For instance, circRILPL1 can be used as a miR-145 sponge to regulate *IGF1R*, thus regulating the growth of bovine myoblasts. However, when cyclophosphamide was injected into the muscle of mice, circRILPL1 could adsorb miR-145 to regulate the downstream pathways, thus promoting muscle regeneration [116]. CircMYBPC1 regulated bovine myoblast differentiation and promoted skeletal muscle regeneration following muscle injury [118]. Overexpression of circEch1 inhibited the proliferation of bovine myoblasts and promoted their differentiation. In vivo studies suggested that circEch1 induced skeletal muscle regeneration [119]. CircCPE promoted the proliferation of bovine myoblasts and inhibited apoptosis and differentiation in vitro. It also attenuated skeletal muscle regeneration in vivo [120]. These data suggest that circRNAs play a critical role in skeletal muscle regeneration in cattle.

### 5.3. Sheep and Goats

Sheep and goats are multi-purpose animals that produce meat, milk, skins, and wool/hair at the beginning of domestication. Recently, Ling et al. performed RNA-seq analysis on the skeletal muscles of Anhui white goats on days 45, 65, 90, 120, and 135 during embryonic development and days 1 and 90 after birth. They identified 9090 novel circRNAs, including 2881 DE-circRNAs. Enrichment analysis suggested that the host genes of these DE-circRNAs are associated with Wnt signaling, AMPK signaling, and other pathways related to muscle development. Notably, they reported that some circRNAs could affect muscle formation through metabolism, regulation of enzyme activity, and biosynthesis [121]. RNA-seq analysis was performed to identify circRNAs expressed in the *longissimus dorsi* of adult Kazakh sheep. Many of the 886 identified circRNAs, including circ776, could interact with miRNAs associated with muscle growth and development [122]. In addition, Li et al. analyzed RNA-seq data from embryonic to newborn goat skeletal muscle. They identified a typical E-box element in the CDR1as promoter that binds the MyoD protein at the beginning of the differentiation of skeletal muscle satellite cells and substantially promotes the transcription of CDR1as. CDR1as could induce myogenesis at least partially through the CDR1as/miR-7/IGF1R regulatory pathway. The authors also proposed that CDA1as may encode functional peptides in skeletal muscle [56].

### 5.4. Poultry

Chicken is one of the most essential components of livestock production. To identify circRNAs that are potentially associated with chicken embryonic skeletal muscle development, Ouyang et al. performed RNA-seq on the leg muscles of female Xinghua (XH) chickens at three developmental time points (11 embryo age (E11), E16, and 1 d post-hatch (P1)). They identified 13,377 circRNAs in embryonic skeletal muscle, including 462 circRNAs that were differentially expressed between different developmental stages [123]. Further analysis suggested circRBFOX2s could promote myoblast proliferation by interacting with miR-206 [123]. CircFGFR2 promotes myoblast proliferation and differentiation by sponging miR-133a-5p and miR-29b-1-5p [124]. CircSVIL was highly expressed in the late stage of embryonic skeletal muscle development, and circSVIL could upregulate the mRNA levels of c-JUN and MEF2C and function as miR-203 sponges. Additionally, circSVIL can promote the proliferation and differentiation of myoblasts and inhibit the functions of miR-203 [125]. CircFAM188B is differentially expressed between broiler chickens and layers during embryonic skeletal muscle development. Functional analysis suggested that circFAM188B promotes proliferation and inhibits differentiation of chicken skeletal muscle satellite cells (SMSCs). CircFAM188B contains an open reading frame and can be translated into a novel protein, circFAM188B-103aa [126]. Additionally, circPPP1R13B promotes chicken SMSCs proliferation and differentiation by targeting miR-9-5p and activating the IGF/PI3K/AKT signaling pathway [127], whereas circTMTC1 inhibits the differentiation of chicken SMSC through the miR-128-3p sponge [128]. MiR-30a-3p reportedly affects the proliferation and differentiation of myoblasts [129,130]. In chicken embryo skeletal muscle, circHIPK3 acts as a sponge for miR-30a-3p, thus promoting the proliferation and differentiation of myoblasts [130]. 

**Table 1 biomolecules-13-00314-t001:** Summary of circRNAs and their functions in skeletal muscle development.

CircRNA	Species	Function	Mechanism ^1^	Cells ^2^	Ref.
CircFgfr2	Pig	Inhibit myoblast proliferation/ promotes differentiation and skeletal muscle regeneration	As miR-133 sponge	Mouse primary myoblasts/C2C12	[37]
CircTUT7	Pig	Regulate HMG20B expression	As miR-30a-3p sponge	/	[106]
Circ_0015885	Pig	Regulate skeletal muscle growth	As miR-23b sponge	/	[107]
CircMYLK4	Pig	Affecting muscle fiber type conversion	/	/	[110]
CircHUWE1	Bovine	Promote myoblast proliferation, Inhibit apoptosis and differentiation	As miR-29b sponge	Myoblasts	[112]
CircSNX29	Bovine	Inhibit proliferation/ promote differentiation	As miR-744 sponge	Primary myoblasts	[113]
CircTTN	Bovine	Promote proliferation and differentiation	As miR-432 sponge	Primary myoblasts	[115]
CircFUT10	Bovine	Inhibit proliferation/induce differentiation	As miR-133a sponge	Myoblasts	[114]
CircFGFR4	Bovine	Promote differentiation/induce apoptosis	As miR-107 sponge	Primary myoblasts	[117]
CircLMO7	Bovine	Regulate cell differentiation and survival	As miR-378a-3p sponge	Myoblasts	[111]
CircINSR	Bovine	Promote myoblasts development	As miR-15/16 sponge	Primary myoblasts	[8]
CircRILPL1	Bovine	Promote muscle regeneration/regulate myoblasts growth	As miR-145 sponge	Primary myoblasts	[116]
CircEch1	Bovine	Inhibit proliferation/promote differentiation/induce regeneration	/	Myoblasts	[119]
CircCPE	Bovine	Promote proliferation/inhibit apoptosis and differentiation/induce regeneration	As miR-138 sponge	Primary myoblasts	[120]
CircMYBPC1	Bovine	Promote skeletal muscle regeneration	Bind MyHC protein	Primary muscle cells	[118]
CDR1as	Goat	Bind to MyoD protein/the transcription of CDR1as/induce myogenesis	As miR-7 sponge	SMSCs	[56]
CircRBFOX2s	Chicken	Promote proliferation	As miR-206 sponge	Myoblasts	[123]
CircFGFR2	Chicken	Promote proliferation and differentiation	As miR-133a-5p/miR-29b-1-5p sponge	DF embryo fibroblast cell line (DF-1) cells/primary myoblasts	[124]
CircSVIL	Chicken	Promote proliferation and differentiation	As miR-203 sponge	Primary myoblasts	[125]
CircFAM188B	Chicken	Promote proliferation and inhibit differentiation	Translate protein	SMSCs	[100]
CircPPP1R13B	Chicken	Promote proliferation and differentiation	As miR-9-5p sponge	SMSCs	[127]
CircTMTC1	Chicken	Inhibit differentiation	As miR-30 a-3p sponge	SMSCs	[128]
CircHIPK3	Chicken	Promote proliferation and differentiation	As miR-30a-3p sponge	Primary myoblasts	[130]

^1^ The regulatory mechanism of this circRNA. ^2^ The cells used for functional and mechanism studies of this circRNA.

## 6. Identification and Functions of CircRNAs in Adipose Tissues of Farm Animals

### 6.1. Pigs

Fat deposition and IMF content in farm animals substantially affect meat yield and quality. To identify circRNAs potentially involved in fat deposition in pigs, Liu et al. isolated preadipocytes from subcutaneous adipose tissue of Chinese Erhualian piglets on day 5 and performed RNA-seq at different differentiation stages (D0, D2, D4, and D8). They identified a total of 8623 circRNAs [131]. Li et al. performed RNA-seq analysis on the subcutaneous adipose tissues of Large White and Chinese Laiwu pigs and identified 29,763 circRNAs, including 275 circRNAs that were differentially expressed between these breeds. Notably, they observed that the targets of circRNA_26852 and circRNA_11897 were enriched in pathways related to adipocyte differentiation and lipid metabolism, suggesting that these two circRNAs may be candidate genes associated with fat deposition [132]. Finally, Zhang et al. investigated the circRNA profile of porcine *longissimus dorsi* with different intramuscular fat content using RNA-seq. A total of 29,732 circRNAs, including 336 DE-circRNAs, were identified. Notably, they identified a new circRNA, circPPARA, mainly expressed in the cytoplasm. CircPPARA can inhibit the proliferation of porcine intramuscular preadipocytes and promote intramuscular fat generation by adsorbing miR-429 and miR-200b (Table 2) [133]. By analyzing the expression of circRNAs in subcutaneous adipose tissue of male pigs, Wang et al. identified 6116 DE-circRNAs between intact and castrated male Huainan pigs, which are highly enriched in metabolism-related pathways. Notably, they reported that circ_0005912 has eight binding sites for miR-181a, and overexpression of miR-181a inhibited the expression of circ_0005912 (Table 2). Meanwhile, testosterone could increase the expression of miR-181a in intramuscular adipocytes and decrease the expression of circ_0005912 in a dose-dependent manner, suggesting circ_0005912 may regulate fat deposition in castrated pigs [134]. 

### 6.2. Bovines

Li et al. profiled circRNA expression in the adipose tissues of calves and adult cattle using Ribo-Zero RNA-Seq. A total of 14,274 candidate circRNAs were identified. They further focused on the differentially and abundantly expressed circRNA-circFUT10. CircFUT10 can promote proliferation and inhibit differentiation of bovine adipocytes by sponging let-7c. Let-7c can target PPARGC1B (PPARg cofactor 1-b), which can regulate the expression of adipogenesis markers PPARγ and C/EBPβ [135]. CircINSR reportedly inhibits preadipocyte adipogenesis by reducing the repression of FOXO1 and EPT1 by sponging miR-15/16. miR-15/16 promotes the expression of adipogenic genes and lipid accumulation in preadipocytes [8]. Meanwhile, circFLT1 also affects bovine fat development. Overexpression of circFLT1 significantly reduces the expression of miR-93 in preadipocytes, which could inhibit adipocyte differentiation. In addition, overexpression of miR-93 could decrease the expression of PPARG, CEBPA, and other adipocyte differentiation markers. These results suggest that circFLT1 promotes adipocyte differentiation by reducing the expression of miR-93 [136] (Table 2). 

Yak is a vital source of meat and milk for Tibetans and other nomadic pastoralists. Zhang et al. explored the dynamic expression of circRNAs during yak-adipocyte differentiation. A total of 7203 novel circRNAs were identified, among which 136 were differentially expressed between days 0 and 12 after differentiation. ceRNA network analysis helped identify six circRNAs potentially related to adipogenesis. These were targeted by adipocyte differentiation-related miRNAs, such as miR-143, miR-378, miR-328, and miR-34a [137]. Huang et al. identified 5141 circRNAs in the adipose tissue of Chinese buffaloes, among which 252 were differentially expressed between young and adult buffaloes. Notably, 34 circRNAs were highly correlated with adipose deposition-related genes, including two circRNAs that are potential regulators of buffalo fat deposition [138].

### 6.3. Sheep and Goats

To identify potential circRNAs associated with fat deposition, Xiao et al. performed RNA-seq and compared gene expression differences (mRNAs, lncRNAs, and circRNAs) between preadipocytes and mature adipocytes in Chinese Small Tail Han sheep. A total of 3480 circRNAs, including 17 DE-circRNAs, were identified. Regulatory network analysis suggested that four circRNAs (circRNA0002331, circRNA0000520, circRNA000297, and circRNA0002909) are potentially associated with adipocyte differentiation. However, their functions and regulatory mechanisms require further investigation [139]. Zhao et al. assessed the IMF content in Aohan fine-wool sheep aged 2, 4, 6, and 12 months and observed significant differences between 2- and 12-month-old sheep. By analyzing RNA-seq data of *longissimus dorsi* muscle from these two age groups, they identified 11,565 candidate circRNAs and 104 DE-circRNAs. The DE-circRNAs have significantly enriched in fat metabolism-related pathways, and circRNA455, circRNA4557, and circRNA2440 may be involved in IMF deposition [140]. The regulatory mechanisms associated with adipogenesis are complex and involve multiple regulatory pathways, transcription factors, and hormones, as well as different types of adipose tissue and different types of adipocytes. To study the potential mechanism of circRNA transformation from brown to white adipose tissue, Zhang et al. performed RNA-seq on the perirenal adipose tissue of goats at three developmental stages (D1, D30, and Y1). Among the 6610 novel circRNAs identified, 61 were differentially expressed. They further constructed a DEcircRNA–miRNA interaction network and identified circ20832_5 (derived from NRG4) and circ3842_1 (circulated from PRKAG2), potentially associated with brown-to-white adipose tissue transformation [141]. These findings suggest that circRNAs are associated with fat deposition and IMF content, providing a reference for further research on sheep fat development.

### 6.4. Poultry

To explore the function of circRNAs in chicken adipocyte differentiation and fat deposition, Zhang et al. performed Ribo-Zero RNA-Seq. They examined the dynamic expression of circRNAs during intramuscular and abdominal adipogenic differentiations. Through ceRNA network analysis, several candidate circRNAs, such as circLCLAT1, circFNDC3AL, circCLEC19A, and circARMH1, potentially influencing adipogenesis in chickens by acting as miRNA sponges, were identified [142]. CircRNA expression profiles during abdominal adipose tissue development in Chinese Gushi chickens suggested that most of the identified circRNAs had IRES elements and exhibited developmental-specific expression patterns. Functional enrichment analysis of the parental genes of DE-circRNAs and ceRNA network analysis suggested that circRNAs could regulate lipid metabolism, adipocyte proliferation, and differentiation in chickens [143]. A recent study reported that 141 circRNAs are differentially expressed between preadipocytes and differentiated adipocytes in Cherry Valley ducks. Through ceRNA analysis, 10 circRNAs potentially involved in duck adipocyte differentiation, including circ-PLXNA1, were identified. Expression analysis suggested circ-PLXNA1 is mainly expressed in the adipose tissue, leg muscles, and the liver. Functional studies revealed that circ-PLXNA1 knockdown could inhibit the differentiation of preadipocytes and may function as a ceRNA to regulate CTNNB1 expression by sponging miR-214 during duck preadipocyte differentiation [144].

**Table 2 biomolecules-13-00314-t002:** Summary of circRNAs and their functions in fat deposition.

CircRNA	Species	Function	Mechanism ^1^	Cells ^2^	Ref.
CircPPARA	Pig	Inhibit proliferation /Promote fat formation	As miR-429/miR-200b sponge	Intramuscular preadipocytes	[133]
Circ_0005912	Pig	Regulate fat deposition	As miR-181a sponge		[134]
CircINSR	Bovine	Reduce fat formation	As miR-15/16 sponge	Preadipocytes	[8]
CircFUT10	Bovine	Promote proliferation/ Inhibit differentiation	ceRNA	Adipocytes	[135]
CircFLT1	Bovine	Promote differentiation /Inhibit proliferation	As miR-93 sponge	Adipocytes/HEK293T/3T3	[136]
Circ-PLXNA1	Duck	Affect differentiation	As miR-214 sponge	Primary preadipocytes	[144]

^1^ The regulatory mechanism of this circRNA. ^2^ The cells used for functional and mechanism studies of this circRNA.

## 7. Conclusions and Perspectives

In this review, we have summarized the current progress on the research of circRNAs in skeletal muscle and adipose development of farm animals (Table 1 and Table 2). The reviewed studies have identified functional circRNAs involved in myogenesis and adipogenesis. Our review helps further understand the mechanisms underlying the regulation of skeletal muscle and adipose development. We propose that circRNAs are promising targets for improving meat yield and quality traits in farm animals. 

However, compared to circRNA studies in humans and mice, knowledge regarding the dynamic expression, biological function, and regulatory mechanism of circRNA in farm animals still needs to be improved. First, millions of circRNAs have been identified, and multiple circRNA databases have been constructed for humans, mice, and other species [145,146,147]. For example, the CircAtlas database contains over one million highly reliable circRNAs across six vertebrate species [148]. However, circRNA databases of farm animals are rare. A comprehensive database that collects circRNAs in all farm animals is required to provide resources for studying animal genetics and breeding. Our group constructed the first open circRNA database from pigs, which are important farm animals [105]. Second, although thousands of circRNAs have been identified in the transcriptomes of farm animals using different software, only a few have been validated at the cellular and molecular levels, and their functions and regulatory mechanisms in skeletal muscle and adipose development still need to be discovered. 

Moreover, most functional studies of candidate circRNAs focus on the ceRNA mechanism. The other regulatory mechanisms of circRNAs in skeletal muscle and adipose development in farm animals are rarely studied. Third, the rapid development of single-cell technologies has provided powerful tools for studying gene expression, epigenetic modification, and the spatial location of cell types at single-cell resolution. CircRNAs are known to exhibit highly specific spatial and temporal expression patterns. Recently, circSC, the first single-cell circRNA database, was constructed to demonstrate the high cell-type specificity of circRNAs at human and mouse resolution [149]. Therefore, single-cell technologies could be used to analyze the expression, function, and location of circRNAs during skeletal muscle and fat development in farm animals. Finally, further functional studies of circRNAs using gene editing tools, such as CRISPR-Cas9 technology, at the molecular, cellular, and individual levels will help elucidate the mechanisms of circRNAs during skeletal muscle and fat development. Fourth, the breeding value of candidate circRNAs in the genetic improvement of meat yield and quality traits should be further evaluated and validated at the population level. The application of circRNAs in animal breeding should be explored further.

## Figures and Tables

**Figure 1 biomolecules-13-00314-f001:**
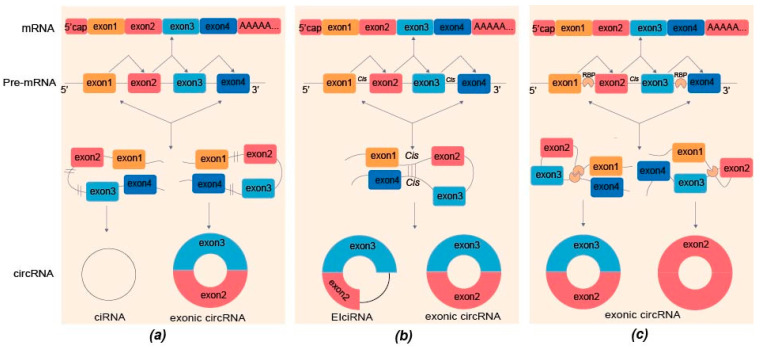
Mechanism of circRNAs biosynthesis. (**a**) Back-splicing: Spliceosomes connect downstream 5′ and upstream 3′ splice sites to form circRNAs by reverse splicing pre-mRNA; ciRNA is generated when the intron escapes the branching lasso during the canonical splicing process of exon circularization. ciRNA, intronic circRNA. (**b**) *Cis*-regulation elements: the *cis*-element makes RNA pairing between introns on either side of the exon easier by reducing the spatial obstruction of back-splicing. There are many RNA pairs in flanking introns and inverted repeat Alu elements, and exon cyclization, depends on the complementary sequence of flanking introns. The Alu element in introns can promote the production of EIciRNA. EIciRNA, exon–intron circRNA. (**c**) Two RNA-binding protein patterns regulate circRNAs biosynthesis. RBP binds to other RBPs or intron sequences to form exonic circRNAs by back-splicing. RBP, RNA binding protein.

**Figure 2 biomolecules-13-00314-f002:**
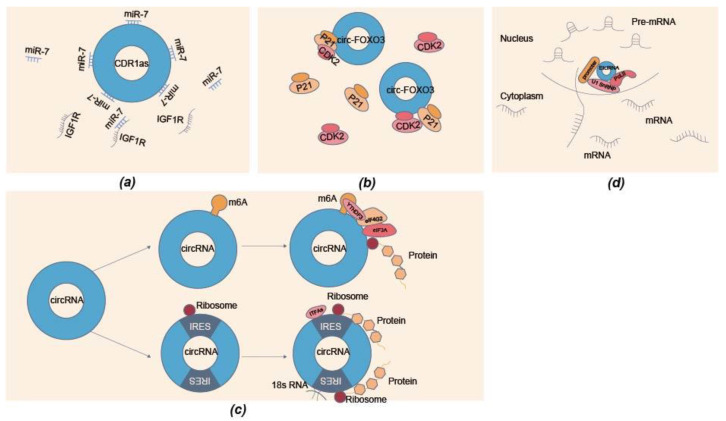
Regulatory mechanisms of circRNAs. (**a**) miRNA sponge: circRNAs can function as miRNA sponges. For example, circRNA CDR1as downregulates insulin-like growth factor 1 receptor (IGF1R) expression by acting as a sponge or inhibitor of miR-7. (**b**) Interactions with proteins: circ-Foxo3 forms ternary complexes with p21 and CDK2. (**c**) Protein translation: circRNAs have the translation potential by relying on IRES and/or m6A-mediated translation. (**d**) Transcription regulation: circRNA can regulate transcription to affect gene expression. For example, EIciRNA, U1 snRNP and Pol II may interact in the promoter region of the parent gene to affect parental gene expression.

## Data Availability

Not applicable.

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
