# Peer review of "Recent Progress on Circular RNAs in the Development of Skeletal Muscle and Adipose Tissues of Farm Animals"

_biomolecules, 2023, doi:10.3390/biom13020314_

Round 1

Reviewer 1 Report

Shanying Yan et al. present a review on circular RNAs. They focus on their functions and mechanisms in adipose tissue and skeletal muscle development of farm animals. Biogenesis and functions of circRNA are described in a general way, and then focus to farm animals to show that circular RNAs could be actors to improve meat yield and quality.

Even if the review is quite exhaustive some clarifications are required. The following points should be addressed. 

1)     The review needs a thorough proofreading to avoid all typos. The style is heavy and I suggest using short sentences. The manuscript will benefit greatly in terms of clarity.

2)     Page 2 line 3: There is a semicolon instead of a point.

3)     Page 2 line 4: What does it mean: “an erroneously spliced mRNA product”. Add some precisions.

4)     Page 2 line 18: The authors write “spaces” instead of “species”.

5)     Page 2 in part 2 The authors could add that there are also intronic-exonic cirRNA

6)     Page 2: “A single gene can produce multiple circRNAs with different lengths and sequences by alternative back-splicing but share the same back-splicing site” Could the authors clarify this sentence?

7)     Figure 1 is wrong. The 5’cap should be bounded to exon1 and the polyA to the exon 3. It is impossible to have a back-splicing between exon1 and 3 because there are no splice sites. This figure must be remodeled.

8)     Page 3: “For example, circRNA CDR1as has approximately 70 binding sites for miR-7 [13], which can downregulate the insulin-like growth factor 1 receptor (IGF1R) expression by acting as a miR-7 sponge or inhibitor “. There is a missense because from the referenced article: “By competitively binding to miR-7 (microRNA 7), CDR1as relieves the downregulation of IGF1R (insulin like growth factor 1 receptor) caused by miR-7 and consequently activates muscle differentiation”

9)     Page 4: “CNEACR is a cardiac necrosis-associated circRNA that binds directly to histone deacetylase (HDAC7) in the cytoplasm, interfering with its entry into the nucleus and leading to the repression of transcription of forkhead box protein A2 (Foxa2), which can affect myocardialnecrosis by binding to the promoter region to repress the receptor-interacting protein kinase 3 (Ripk3) gene [66] » . There is a missense because from the referenced article: “Mechanistically, CNEACR directly binds to histone deacetylase (HDAC7) in the cytoplasm and interferes its nuclear entry. This leads to attenuation of HDAC7-dependent suppression of forkhead box protein A2 (Foxa2) transcription, which can repress receptor-interacting protein kinase 3 (Ripk3) gene by binding to its promoter region.”

10)   Page 4 line 24: The authors should remove “And”

11)   Page 4 first paragraph of 3.3: The authors should add some details about ribosomal recruitment to the IRES.

12)   Figure 2:
Figure 2a There is a bad representation of miR-7 and IGF1R. RNA IGFR has to be single strand and miR-7 should be represented as a short single strand RNA.
Figure 2c The representation is wrong because it is ITAFs that recruit ribosome and not the contrary.
Figure 2d: What is ATG ? mRNA is not double strand. The figure is not understandable some precision need to be add.

13)   Page 6 line 36 There is a missing space between DE circRNAs

14)   Table 1 The writing need to be homogenize, either capitalize all the circs or lowercase them.

15)   Page 9 line 14 CircPPARA placed at the beginning of the sentence need to have a capital letter.

Author Response

Point-by-Point Response to Referee 1’ Comment

Shanying Yan et al. present a review on circular RNAs. They focus on their functions and mechanisms in adipose tissue and skeletal muscle development of farm animals. Biogenesis and functions of circRNA are described in a general way, and then focus to farm animals to show that circular RNAs could be actors to improve meat yield and quality.

Even if the review is quite exhaustive some clarifications are required. The following points should be addressed.

Response: We are truly grateful to you and the other reviewer for providing constructive and thoughtful comments, which greatly help us to improve this manuscript. We have addressed the specific points made by you and the other reviewer. We have incorporated your suggestions into the revised manuscript, as detailed in the point-by-point response below.

1. The review needs a thorough proofreading to avoid all typos. The style is heavy and I suggest using short sentences. The manuscript will benefit greatly in terms of clarity.

Response: Thank you very much for your suggestion. We have revised it again carefully and invited a native speaking editor to help us improving the writing of the manuscript. Please read the article for details.

2. Page 2 line 3: There is a semicolon instead of a point.

Response: We apologize for this error. We have revised it accordingly. Please see Page 2 line 3 in the revised manuscript.

3. Page 2 line 4: What does it mean: “an erroneously spliced mRNA product”. Add some precisions.

Response: Thank you for your constructive comments. We have added more details about it in the revised manuscript. Please see Page 2 line 5.

4. Page 2 line 18: The authors write “spaces” instead of “species”.

Response: We are sorry for the typo error and have revised it accordingly. Please see Page 2 line 21 in the revised manuscript.

5. Page 2 in part 2 The authors could add that there are also intronic-exonic circRNA

Response: Thank you very much for your suggestion. We have added more details about intronic-exonic circRNA in the revised manuscript. Please see page 2, line 40.

6. Page 2: “A single gene can produce multiple circRNAs with different lengths and sequences by alternative back-splicing but share the same back-splicing site” Could the authors clarify this sentence?

Response: Thank you very much. We have clarified this sentence in the revised manuscript. Please see Page 2 lines 44-46.

7. Figure 1 is wrong. The 5’cap should be bounded to exon1 and the polyA to the exon 3. It is impossible to have a back-splicing between exon1 and 3 because there are no splice sites. This figure must be remodeled.

Response: Thank you for your kind reminder, we have realized this error and have modified the figure, please see the article for details.

8. Page 3: “For example, circRNA CDR1as has approximately 70 binding sites for miR-7 [13], which can downregulate the insulin-like growth factor 1 receptor (IGF1R) expression by acting as a miR-7 sponge or inhibitor “. There is a missense because from the referenced article: “By competitively binding to miR-7 (microRNA 7), CDR1as relieves the downregulation of IGF1R (insulin like growth factor 1 receptor) caused by miR-7 and consequently activates muscle differentiation”

Response: We are sorry about this and appreciate your comments! We have modified it to “For example, circRNA CDR1as has approximately 70 binding sites for miR-7 [13], which can upregulate the insulin-like growth factor 1 receptor (IGF1R) expression by acting as a miR-7 sponge (Figure 2a), thereby activating muscle differentiation [56]”. Please see Page 3 line 41 to Page 4 line 2 in the revised manuscript.

9. Page 4: “CNEACR is a cardiac necrosis-associated circRNA that binds directly to histone deacetylase (HDAC7) in the cytoplasm, interfering with its entry into the nucleus and leading to the repression of transcription of forkhead box protein A2 (Foxa2), which can affect myocardialnecrosis by binding to the promoter region to repress the receptor-interacting protein kinase 3 (Ripk3) gene [66] ». There is a missense because from the referenced article: “Mechanistically, CNEACR directly binds to histone deacetylase (HDAC7) in the cytoplasm and interferes its nuclear entry. This leads to attenuation of HDAC7-dependent suppression of forkhead box protein A2 (Foxa2) transcription, which can repress receptor-interacting protein kinase 3 (Ripk3) gene by binding to its promoter region.”

Response: Thank you for your reminder. We have modified this sentence to “CNEACR is a cardiac necrosis associated circRNA which binds directly to histone deacetylase 7 (HDAC7) in the cytoplasm and interferes its nuclear entry. This leads to reduced transcriptional inhibition of forkhead box A2 (Foxa2), thereby upregulating receptor interacting protein kinase 3 (Ripk3) gene to affect myocardial necrosis”. Please see Page 4 line 27 in the revised manuscript.

10. Page 4 line 24: The authors should remove “And”

Response: We have removed this word. Please see Page 4 line 31 in the revised manuscript.

11. Page 4 first paragraph of 3.3: The authors should add some details about ribosomal recruitment to the IRES.

Response: Thank you for your constructive comments. More details about ribosomal recruitment to the IRES have been added in the revised manuscript. Please see Page 4 line 47 to Page 5 line 3.

12. Figure 2:
Figure 2a There is a bad representation of miR-7 and IGF1R. RNA IGFR has to be single strand and miR-7 should be represented as a short single strand RNA.
Figure 2c The representation is wrong because it is ITAFs that recruit ribosome and not the contrary.
Figure 2d: What is ATG? mRNA is not double strand. The figure is not understandable some precision need to be add.

Response: Thanks for the constructive advice. We have realized these errors and have modified the figure accordingly. Please see revised Figure 2 in the new version.

13. Page 6 line 36 There is a missing space between DE circRNAs

Response: Thank you for your reminder. We have revised the manuscript thoroughly and uniformly use DE-circRNA as the abbreviation of “Differentially expressed circRNAs”.

14. Table 1 The writing need to be homogenize, either capitalize all the circs or lowercase them.

Response: These typo error has been revised. Thank you very much.

15. Page 11 line 18 CircPPARA placed at the beginning of the sentence need to have a capital letter.

Response: Thank you for your kind reminder. We have revised this. Please see Page 11 line 6 in the revised manuscript.

Reviewer 2 Report

Dear Authors,

The manuscript is interesting and well-written. It introduces some of the order in the knowledge about circRNAs, which play a significant role during gene expression and in the protein-protein interaction. Below I have some suggestions about lacks and corrections to this manuscript:

Figure 1 is not well described, is a lack of development in the shortcuts, please complete it.

subsection regulatory mechanism of circRNAs should be divided into two: one when cirCRNA play a role as separate molecules regulate gene expression and the second when it acts combined with the other molecules,

The "circRNA translation" title is misleading because circRNAs are not translated, but they mediate in protein translation combined with m6A or IRES, which are responsible for direct protein translation. In the subsection transcriptional regulators, the title is also misleading because this part considers transcription of circRNA from other genes, so it is not the regulatory role of circRNA, but the part on circRNA biogenesis and should be moved to this subsection. From line" Simultaneously, ..." is given the real role of circRNA in transcription regulation. This subsection should be rearranged.

Figure 2  - "circRNA can translate protein" is a little misleading please exchange that it can influence or mediate the protein translation

b) point is not clear protein-protein interaction is not precise, maybe the complex of circRNA with a protein complex

Besides, after this subsection, the authors can add one section about identifying circRNA in the laboratory and how their mechanisms or biogenesis are identified by using which laboratory tool? it completes this review.

Tables 1 and 2 should add the footnotes, what does mean "cells" and "mechanisms", and one sentence.

overall is worth it that the authors are participating in the preparation of circRNA databases, especially for farm animals, because the information on the ncRNA regulation is still scarce, especially in farm animals.

Author Response

Point-by-Point Response to Referee 2’ Comment

Dear Authors,

The manuscript is interesting and well-written. It introduces some of the order in the knowledge about circRNAs, which play a significant role during gene expression and in the protein-protein interaction. Below I have some suggestions about lacks and corrections to this manuscript:

Response: Thank you very much for the positive comments about our manuscript. We have addressed all the comments below and modified the manuscript accordingly.

Figure 1 is not well described, is a lack of development in the shortcuts, please complete it.

Response: Thanks to your comments. More details about Figure 1 have been added in the legend of Figure 1. Please see Page 3 Line 13-23 in the revised manuscript.

subsection regulatory mechanism of circRNAs should be divided into two: one when circRNA play a role as separate molecules regulate gene expression and the second when it acts combined with the other molecules,

Response: Thank you very much. We totally agree your comments about the regulatory mechanism of circRNAs. circRNAs can directly regulate gene expression or interact with other molecules to function across a myriad of biological contexts. In our understanding, these regulatory mechanisms could not be distinguished from each other completely. To make this section clear, more details about the regulatory mechanism have been added as your suggestion. Plesase see revised section 3 in the revised manuscript.

The "circRNA translation" title is misleading because circRNAs are not translated, but they mediate in protein translation combined with m6A or IRES, which are responsible for direct protein translation. In the subsection transcriptional regulators, the title is also misleading because this part considers transcription of circRNA from other genes, so it is not the regulatory role of circRNA, but the part on circRNA biogenesis and should be moved to this subsection. From line" Simultaneously, ..." is given the real role of circRNA in transcription regulation. This subsection should be rearranged.

Response: Thank you for your constructive comments. Although most circRNAs are not translated, however, with deep research, increasingly more circRNA have been reported to have peptide- or protein-coding potential. For example, Circ-ZNF609 could be translated into a protein in a splicing-dependent and cap-independent manner to regulate myogenesis. So in this section, we focus on to review the current progresses about the translation potential of circRNAs and their potential mechanisms. To avoid confusion, we changed the title of this section to “3.3       Translation potential of circRNAs”. Meanwhile, according to your suggestion, we have rearranged this subsection for clearly clarifying this. Please see revised section 3.4 in the manuscript.

Figure 2-"circRNA can translate protein" is a little misleading please exchange that it can influence or mediate the protein translation

Response: Thanks to your comments, we have modified it to “have the translation potential”. Please see Page 6 line 7 in the revised manuscript.

  1. b) point is not clear protein-protein interaction is not precise, maybe the complex of circRNA with a protein complex

Response: Thanks to your comments, we have modified it to “Interactions with proteins”. Please see Page 6 line 5 in the revised manuscript.

Besides, after this subsection, the authors can add one section about identifying circRNA in the laboratory and how their mechanisms or biogenesis are identified by using which laboratory tool? it completes this review.

Response: It’s really a great suggestion. We have added a new section “4. Methodologies for circRNA identification” in the revised manuscript. In this section, we reviewed recent progresses about circRNA identification using bioinformatics methods and in wet experiments. Please see new section 4 of the revised manuscript.

Tables 1 and 2 should add the footnotes, what does mean "cells" and "mechanisms", and one sentence.

Response: Thank you very much. Footnotes have been added to explain "cells" and "mechanisms" in the revised manuscript.

overall is worth it that the authors are participating in the preparation of circRNA databases, especially for farm animals, because the information on the ncRNA regulation is still scarce, especially in farm animals.

Response: Thank you again for your valuable comments.

Round 2

Reviewer 2 Report

Dear Authors, I see that all my suggestions were considered, so I propose to accept this manuscript in its present form. I do not have additional comments.